# SPLAT THE NET: RADIANCE FIELDS WITH SPLATTABLE NEURAL PRIMITIVES

**Xilong Zhou\*, Bao-Huy Nguyen\*, Loïc Magne, Vladislav Golyanik,**
**Thomas Leimkühler, Christian Theobalt**
`*Co-first authors`

Max Planck Institute for Informatics
`{xzhou,bnguyen}@mpi-inf.mpg.de`

## ABSTRACT

Radiance fields have emerged as a predominant representation for modeling 3D scene appearance. Neural formulations such as Neural Radiance Fields provide high expressivity but require costly ray marching for rendering, whereas primitive-based methods such as 3D Gaussian Splatting offer real-time efficiency through splatting, yet at the expense of representational power. Inspired by advances in both these directions, we introduce *splattable neural primitives*, a new volumetric representation that reconciles the expressivity of neural models with the efficiency of primitive-based splatting. Each primitive encodes a bounded neural density field parameterized by a shallow neural network. Our formulation admits an exact analytical solution for line integrals, enabling efficient computation of perspectively accurate splatting kernels. As a result, our representation supports integration along view rays without the need for costly ray marching. The primitives flexibly adapt to scene geometry and, being larger than prior analytic primitives, reduce the number required per scene. On novel-view synthesis benchmarks, our approach matches the quality and speed of 3D Gaussian Splatting while using $10\times$ fewer primitives and $6\times$ fewer parameters. These advantages arise directly from the representation itself, without reliance on complex control or adaptation frameworks. The project page is `https://vcai.mpi-inf.mpg.de/projects/SplatNet/`.

## 1 INTRODUCTION

Radiance fields have become a predominant representation for modeling 3D scene appearance. Unlike surface-based approaches, their volumetric formulation is compatible with the gradient-based optimization routines employed during training from multi-view images. *Neural* representations, particularly Neural Radiance Fields introduced by Mildenhall et al. (2020), offer unprecedented expressivity in encoding radiance fields. However, rendering an image from a volumetric scene representation is generally challenging: Volume rendering (Kajiya & Von Herzen, 1984) requires the computation of costly integrals along view rays, typically solved using quadrature methods such as *ray marching* (Max, 1995). As a remedy, *primitive*-based representations have emerged as an efficient alternative. Popularized by 3D Gaussian Splatting (3DGS) (Kerbl et al., 2023), these approaches model radiance fields using a mixture of simple volumetric functions. The key to high rendering efficiency lies in the observation that these primitives can be easily projected onto the image plane, where they become 2D kernels that can be efficiently *splatted*. A prime example is the 3D Gaussian primitive used in 3DGS, which reduces to a 2D Gaussian splatting kernel (Zwicker et al., 2001). Recently, a variety of functions have been explored as primitives (Mai et al., 2025; von Lützow & Nießner, 2025; Hamdi et al., 2024; Held et al., 2025; Huang et al., 2024), all relying on relatively simple analytical formulations, which are widely considered essential for enabling efficient conversion into view-dependent splatting kernels.

These developments have led to a prevalent, somewhat dichotomous view of radiance field representations: Neural representations are expressive but come with the high cost of ray marching for rendering, whereas primitive-based representations, though simpler and less expressive, offer more

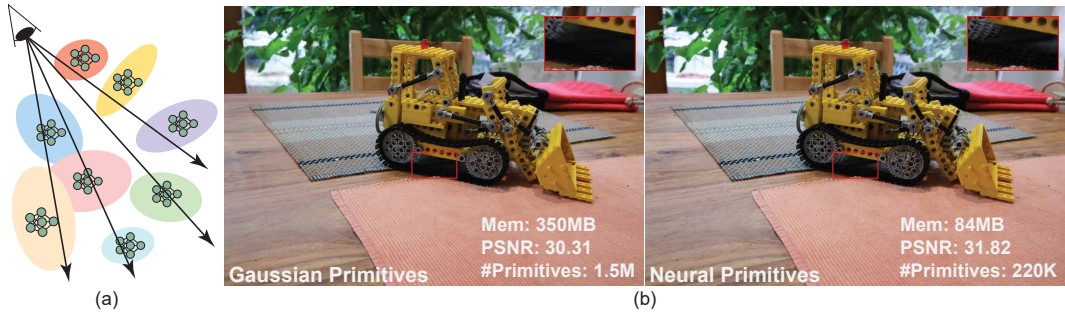

(a)                                                                                      (b)

**Figure 1:** (a) Overview of *volumetric splattable neural primitives*. Each primitive is spatially bounded by an ellipsoid, and its density is parameterized as a shallow neural network. (b) A real scene rendered using Gaussian primitives (left) and neural primitives (right). Our method achieves comparable PSNR to the Gaussian representation but with fewer primitives, highlighting the expressivity of neural primitives.

efficient rendering through splatting. We challenge this common wisdom by introducing *splattable neural primitives*, offering both expressivity and real-time efficiency; see Fig. 1 for an overview.

The central design ingredient of our primitives is a neural volumetric density field. Its density distribution is parameterized by a shallow yet expressive neural network and is spatially bounded by an ellipsoid. This formulation admits an exact analytical solution for line integrals (Subr, 2021; Lloyd et al., 2020), which enables efficient computation of a perspectively accurate image-space footprint, i.e., a splatting kernel for rendering. Despite its neural representation, this enables integration of the density field along each pixel's view ray without the need for costly ray marching. Our primitives flexibly adapt to scene geometry and, being typically larger than the analytic primitives employed in recent work, reduce the total number needed to represent a scene. This yields a highly favorable trade-off among quality, performance, and compactness in novel-view synthesis: We match the quality and speed of 3DGS while using $10\times$ fewer primitives and $6\times$ fewer parameters. Crucially, these advantages result from the design of our representation itself, without requiring complex control or adaptation frameworks (Mallick et al., 2024; Fan et al., 2024). In summary, our contributions are:

- A taxonomy of radiance field representations, highlighting a dichotomy between neural and splatting-based approaches (Sec. 2.1).

- A novel volumetric representation based on splattable neural primitives, bridging the gap between and leveraging the benefits of both neural and primitive-based approaches (Sec. 3).

- The application of the representation to novel-view synthesis, validating its practical effectiveness and efficiency: Neural primitives achieve real-time rendering speed and produce result quality comparable to 3DGS with a smaller memory budget (Sec. 4).

## 2 BACKGROUND AND RELATED WORK

### 2.1 RADIANCE FIELD REPRESENTATION AND RENDERING

Radiance fields represent the appearance of a 3D scene via a function $F_\theta : (\mathbf{x}, \mathbf{d}) \to (\sigma, \mathbf{c})$, which maps a spatial location $\mathbf{x} \in \mathbb{R}^3$ and view direction $\mathbf{d} \in \mathbb{S}^2$ to a volumetric density $\sigma \in \mathbb{R}_+$ and an RGB color $\mathbf{c} \in \mathbb{R}^3$. Synthesizing an image from a radiance field involves emission–absorption volume rendering (Kajiya & Von Herzen, 1984) along view rays $\mathbf{r}(t) = \mathbf{o} + t\mathbf{d}$, where $\mathbf{o} \in \mathbb{R}^3$ denotes the camera center, and $t \in \mathbb{R}_+$ parameterizes the ray. Specifically, each RGB pixel color $C \in \mathbb{R}^3$ is computed by evaluating the radiance field along its corresponding view ray:

$$C(\mathbf{r}) = \int_{t_n}^{t_f} \exp\left(-\int_{t_n}^{t} \sigma\left(\mathbf{r}(s)\right) \mathrm{d}s\right) \sigma\left(\mathbf{r}(t)\right) \mathbf{c}\left(\mathbf{r}(t), \mathbf{d}\right) \mathrm{d}t, \tag{1}$$

where $t_n$ and $t_f$ are near and far bounds, respectively. Recent years have seen considerable research devoted to the foundational question of how to best represent $F_\theta$. Representations can be arranged along an *atomicity scale*, from monolithic models that entangle all components in a single structure to modular formulations made up of many simple, spatially localized primitives (horizontal axis in

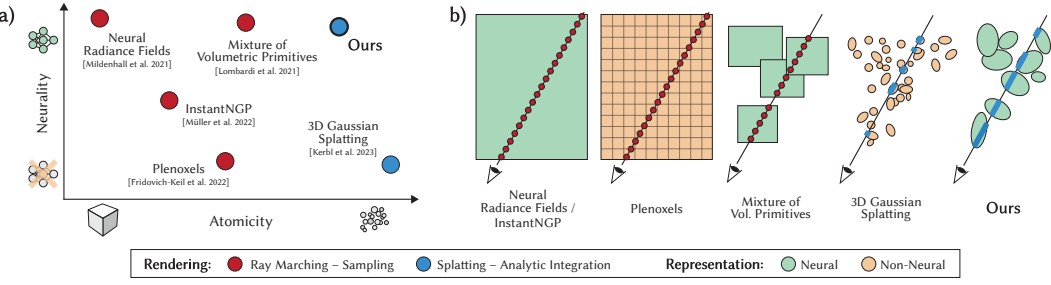

**Figure 2:** Positioning of our work relative to hallmark radiance field representations. *a)* Overview of representations organized along two central design dimensions: Atomicity (horizontal axis), spanning from monolithic (left) to distributed (right) representations; Neurality (vertical axis), ranging from non-neural (bottom) to fully neural (top) approaches. Dot color indicates the supported rendering algorithm. *b)* Illustration of the rendering algorithms associated with each representation. Our method is the only neural, primitive-based model that supports efficient splatting for rendering—thereby eliminating the need for costly ray marching—while retaining the flexibility of a neural design.

Fig. 2a). In the following paragraphs, we briefly review the literature on radiance field representations along this atomicity scale, from monolithic to modular.

Neural Radiance Fields (NeRFs) (Mildenhall et al., 2020) represent $F_\theta$ using a neural network. Numerous follow-up works (Barron et al., 2021; 2022; Martin-Brualla et al., 2021) extended and enhanced its capabilities. However, rendering an image from it involves ray marching, that is, discretizing Eq. 1 into a finite sum of samples along each ray (Max, 1995) (red dots in Fig. 2b):

$$C(\mathbf{r}) \approx \sum_{i=1}^{N} \exp\left(-\sum_{j=1}^{i-1} \sigma_j \delta_j\right) \left(1 - \exp(-\sigma_i)\delta_i\right) \mathbf{c}_i, \tag{2}$$

where $\delta_i$ is the distance between adjacent samples. This rendering process is computationally expensive, as each sample entails a forward pass through the representation network. The pursuit of efficiency has motivated a shift from monolithic to more distributed, explicit representations. Prominent directions include the use of grids (Fridovich-Keil et al., 2022; Barron et al., 2023), volumetric meshes (Govindarajan et al., 2025), and neuro-explicit structures (Müller et al., 2022; Chan et al., 2022; Reiser et al., 2021), all seeking trade-offs between computational cost and memory. Sparse representations allow skipping of empty space (Lombardi et al., 2021; Liu et al., 2020; Yu et al., 2021), reducing the number of samples during ray marching.

Taking a radically different approach by pushing atomicity to the extreme, 3D Gaussian Splatting (3DGS) (Kerbl et al., 2023) represents $F_\theta$ as an unstructured mixture of up to millions of 3D primitive functions. Each primitive, $P_i$, is specified by a small set of parameters that determine its density distribution, $\sigma_i(\mathbf{x})$, along with appearance parameters that capture its view-dependent color, $\mathbf{c}_i(\mathbf{r})$. Rendering an image from this representation can be achieved through splatting, a two-step process: In the first step, $\sigma_i$ is projected to the image plane by integrating along the view ray $\mathbf{r}$ (blue lines in Fig. 2b), yielding a 2D opacity kernel

$$\alpha_i(\mathbf{r}) = 1 - \exp\left(-\int_{-\infty}^{\infty} \sigma_i\left(\mathbf{r}(t)\right) \mathrm{d}t\right). \tag{3}$$

In the second step, Eq. 1 simplifies to highly efficient alpha blending of the 2D kernels:

$$C(\mathbf{r}) \approx \sum_{i \in \mathcal{N}(\mathbf{r})} \mathbf{c}_i(\mathbf{r})\alpha_i(\mathbf{r}) \prod_{j=1}^{i-1} \left(1 - \alpha_j(\mathbf{r})\right), \tag{4}$$

where $\mathcal{N}$ represents the indices of the depth-sorted primitives intersected by the view ray. Evaluating Eq. 3 is generally non-trivial, but for 3D Gaussian primitives $P_i$, the footprint $\alpha_i$ simplifies to a 2D Gaussian kernel under reasonable assumptions (Zwicker et al., 2001; Celarek et al., 2025), enabling high rendering speed and supporting high-quality radiance fields with millions of Gaussians in 3DGS.

The 3D Gaussian is not the only primitive allowing efficient (approximate) conversion into a 2D splatting kernel $\alpha$. Recent work has explored a variety of primitive shapes (Mai et al., 2025; von

Lützow & Nießner, 2025; Hamdi et al., 2024; Held et al., 2025; Chen et al., 2024; Li et al., 2025; Gu et al., 2024; Hamdi et al., 2024; Talegaonkar et al., 2025; Liu et al., 2025), all relying on hand-crafted *analytic* kernels for efficient evaluation of Eq. 3. In contrast, our representation introduces splattable *neural* primitives, greatly increasing modeling flexibility. Neural components have also been used in the context of primitive splatting, for example, by injecting structure into the representation (Lu et al., 2024) or enforcing spatial regularization (Mihajlovic et al., 2024), yet these methods ultimately splat Gaussian functions. To the best of our knowledge, we are the first to represent the volumetric kernel itself – i.e., the density distribution – as a neural network, making the primitive *fundamentally neural* rather than merely Gaussian with neural augmentations.

## 2.2 Integration with Neural Networks

Estimating integrals is common in visual computing. Feed-forward neural networks trained on the integrand can sometimes perform this task effectively. A notable class in this context is shallow neural networks with one hidden layer, which remain universal function approximators (Cybenko, 1989) and can be integrated in closed form (Yan et al., 2013; Zhe-Zhao et al., 2006; Lloyd et al., 2020; Subr, 2021). Our approach builds on this insight by modeling neural primitives that support closed-form integration along view rays. In contrast, deep neural networks have also been applied to integral computation via derivative graphs (Lindell et al., 2021; Teichert et al., 2019), but their high evaluation cost and difficulty in producing consistent integrals along arbitrary rays remain challenges.

## 3 Method

In this section, we first introduce our neural representation (Sec. 3.1), before explaining how to render images using this representation (Sec. 3.2). Finally, we discuss implementation details including our population control strategy, network design, and training protocol (Sec. 3.3).

## 3.1 Representation

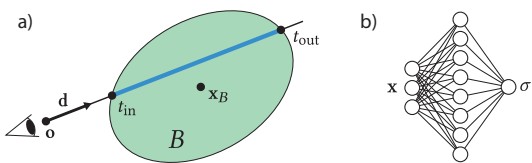

We parameterize $F_\theta$ as a mixture of volumetric primitives $\{P_i\}$. Each primitive occupies a volume bounded by an ellipsoid (Mai et al., 2025), which we denote as $B$ (Fig. 3a). This ellipsoid is defined by a center $\mathbf{x}_B \in \mathbb{R}^3$, a scaling vector $\mathbf{s}_B \in \mathbb{R}^3$ along its principal axes, and a rotation quaternion $\mathbf{q}_B \in \mathbb{R}^4$. In accordance with the radiance field formalism, each primitive must define a spatially vary-

**Figure 3:** *a)* Geometry of our representation for a single primitive. Analytic splatting kernels are computed by performing closed-form integration of a neural density field (green shape) along view rays (blue line). *b)* Architecture of our neural density field. Density $\sigma$ is a function of 3D spatial position $\mathbf{x}$.

ing density $\sigma$ and a view-dependent color $\mathbf{c}$, described next. We define a density field $\sigma(\mathbf{x}) : B \to \mathbb{R}$ within the volume of the ellipsoid as

$$\sigma(\mathbf{x}) = f_\sigma \left( \frac{\mathbf{x} - \mathbf{x}_B}{\|\mathbf{s}_B\|_\infty} \right), \tag{5}$$

where $f_\sigma$ is a shallow neural network with one hidden layer of width $N_\sigma$ and periodic activation (Sitzmann et al., 2020) (Fig. 3b):

$$f_\sigma(\mathbf{x}) = W_2 \left( \cos \left( \omega_0 \left( W_1 \left( \mathbf{x} \right) + \mathbf{b}_1 \right) \right) \right) + \mathbf{b}_2. \tag{6}$$

Here, $W_1 \in \mathbb{R}^{N_\sigma \times 3}$ and $W_2 \in \mathbb{R}^{1 \times N_\sigma}$ are weight matrices, while $\mathbf{b}_1 \in \mathbb{R}^{N_\sigma}$ and $\mathbf{b}_2 \in \mathbb{R}$ are biases. Similar to Sitzmann et al. (2020), we use a fixed boosting frequency $\omega_0$, which yields a stable initialization. The network structure of Eq. 6 admits an interpretation analogous to a Fourier series, where $W_1$ and $\mathbf{b}_1$ correspond to frequencies and phases, and $W_2$ and $\mathbf{b}_2$ are amplitudes and offsets. The normalization by $\mathbf{x}_B$ and $\mathbf{s}_B$ in Eq. 5 ensures that $f_\sigma$ operates on a centered and uniformly scaled domain. In the appendix, we also show proof-of-concept extensions of this model to higher-dimensional inputs, including time, which can be easily incorporated into our model by augmenting the network's input dimensions. To represent view-dependent color, we adopt the Spherical Harmonics basis.

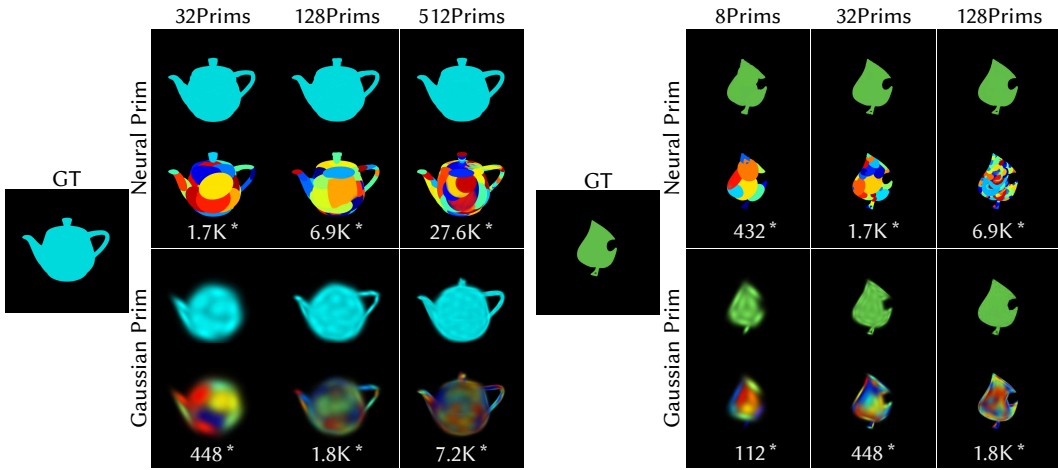

**Figure 4:** Demonstration of the expressivity of the proposed neural density field. We train both neural and Gaussian primitives on the *teapot* and *leaf* datasets using different numbers of primitives. For each example, we visualize the reconstructed density field and color-coded primitives, illustrating how these ellipsoid-bounded neural primitives are deformed to represent complex structures. $*$ denotes the total number of parameters.

## 3.2 RENDERING

Images of our representation are rendered using an efficient splatting-based approach. Specifically, for each primitive bounding ellipsoid $B$ intersected by a view ray $\mathbf{r}(t)$, we compute the entry and exit distances, $t_{\text{in}}$ and $t_{\text{out}}$, along the ray via an analytic line–ellipsoid intersection. To obtain a splatting kernel via Eq. 3, the crucial step is to evaluate the density integral along the view ray (blue line in Fig. 3a):

$$\hat{\alpha}(t_{\text{in}}, t_{\text{out}}, \mathbf{o}, \mathbf{d}) \coloneqq \int_{t_{\text{in}}}^{t_{\text{out}}} \sigma\left(\mathbf{o} + t\mathbf{d}\right) \mathrm{d}t = S\left(t_{\text{out}}; \mathbf{o}, \mathbf{d}\right) - S\left(t_{\text{in}}; \mathbf{o}, \mathbf{d}\right), \tag{7}$$

where $S(t; \mathbf{o}, \mathbf{d})$ denotes the antiderivative with respect to $t$ of the function $t \mapsto \sigma(\mathbf{o} + t\mathbf{d})$, which depends parametrically on $\mathbf{o}$ and $\mathbf{d}$. The equality follows from the fundamental theorem of calculus. Based on recent findings (Lloyd et al., 2020; Subr, 2021), we derive a closed-form antiderivative for our density field:

$$S(t; \mathbf{o}, \mathbf{d}) = [W_2 \oslash (\omega_0 \cdot W_1(\mathbf{d}))] \sin\left(\omega_0\left(t \cdot W_1(\mathbf{d}) + W_1(\mathbf{o}) + \mathbf{b}_1\right)\right) + t \cdot \mathbf{b}_2, \tag{8}$$

where $\oslash$ denotes elementwise division. Incorporating Eq. 3 with the previous derivations yields the final splatting kernel

$$\alpha(\mathbf{r}) = 1 - \exp\left(-\max\left(0, \hat{\alpha}\left(t_{\text{in}}, t_{\text{out}}, \mathbf{o}, \mathbf{d}\right)\right)\right), \tag{9}$$

where the additional clamping to zero ensures nonnegative accumulated density. The final pixel color is determined using front-to-back compositing per Eq. 4.

**Discussion** We emphasize the efficiency of evaluating the splatting kernel via Eq. 9, which computes a closed-form integral along arbitrary view rays through the neural density field, thereby avoiding the computational cost of ray marching. In contrast to splatting-based Gaussian rendering, relying on an affine approximation of the projection operator (Heckbert, 1989; Zwicker et al., 2004), our method yields perspectively accurate results. Note that the density $\sigma$ in Eq. 5 is never evaluated directly, neither during training nor during view synthesis. Instead, all computations operate directly on its antiderivative $S$. Yet, in contrast to a light-field-style approach that directly regresses integrated appearance (Sitzmann et al., 2021), our method achieves multi-view consistency by construction. A detailed analysis of the computational cost of the integration procedure is provided in Appendix A.

## 3.3 IMPLEMENTATION DETAILS

**Primitives** We initialize $W_1 \sim \mathcal{U}\left(-1/3, 1/3\right)$ and $W_2 \sim \mathcal{U}\left(-\sqrt{6/N_\sigma}/\omega_0, \sqrt{6/N_\sigma}/\omega_0\right)$ following Sitzmann et al. (2020). We set the number of hidden neurons $N_\sigma$ to 8 and the frequency

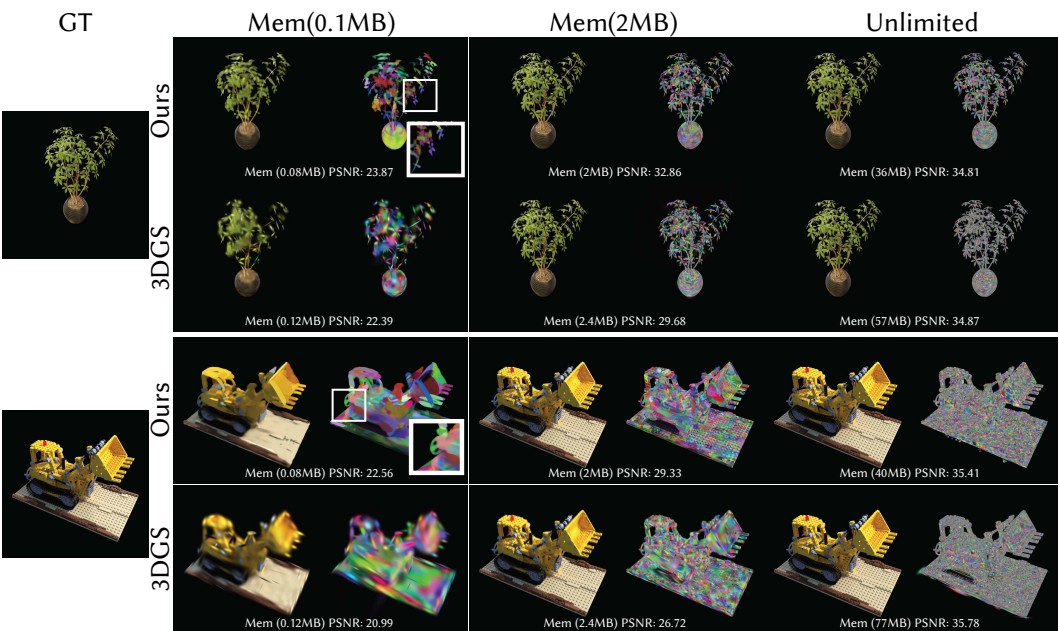

**Figure 5:** Comparison of our method against 3DGS on the synthetic dataset under different memory constraints.

multiplier $\omega_0$ to 30. Similar to 3DGS, we employ four bands of Spherical Harmonics coefficients for color representation. Each neural primitive in our system consists of 99 parameters in total, around $1.6\times$ more than Gaussian primitives used in 3DGS. We provide a detailed analysis of network configurations in Sec. 5.

**Population Control**  Population control is a key factor to the success of primitive-based methods. However, the 3DGS densification strategy is incompatible with neural primitive representations. To address this, we introduce a simple yet effective densification strategy. Unlike 3DGS, which uses the gradients of primitive screen-space locations as the criterion for densification, our approach relies on the gradient magnitude of the network weights. Similar to 3DGS, we duplicate or split primitives when this gradient exceeds a threshold. Primitives with low gradients are pruned. We do not use any opacity resetting.

**Training**  We follow the same loss function as 3DGS, and introduce a geometric regularization term to penalize the extreme anisotropy in primitive shapes by minimizing the standard deviation of the components of the scale vector $\mathbf{s}_B$. The effectiveness of this regularization is demonstrated in Sec. 5. We implement all frameworks in PyTorch (Paszke et al., 2017) and CUDA. All models are trained on a single NVIDIA A40 GPU and evaluated on an NVIDIA RTX 4090 for performance analysis. Due to the complex optimization landscape of neural fields, the convergence of our representation is slower than a Gaussian-based one. We therefore extend training to $100k$ iterations. Additional training details are provided in Appendix B.

## 4  EVALUATION

In this section, we perform a comprehensive evaluation of neural primitives on novel-view synthesis tasks. We first demonstrate the expressivity of the neural density field (Sec. 4.1). We then perform quantitative and qualitative analysis on synthetic datasets (Sec. 4.2) and real datasets (Sec. 4.3) using standard evaluation metrics: Peak Signal-to-Noise Ratio (PSNR), Structural Similarity Index Measure (SSIM) (Wang et al., 2004), and Learned Perceptual Image Patch Similarity (LPIPS) (Zhang et al., 2018). Please refer to Appendix C for additional results.

**Table 1:** Quantitative comparison of our method against 3DGS on the Synthetic NeRF dataset under different memory budgets. We evaluate image quality using three standard metrics: LPIPS, PSNR, and SSIM.

| Mem (MB) | 0.1 | | 0.4 | | 1.0 | | 2.0 | | 4.0 | | Unlimited | |
|---|---|---|---|---|---|---|---|---|---|---|---|---|
| Method | 3DGS | **Ours** | 3DGS | **Ours** | 3DGS | **Ours** | 3DGS | **Ours** | 3DGS | **Ours** | 3DGS | **Ours** |
| PSNR↑ | 23.1 | 24.7 | 25.6 | 27.6 | 27.2 | 28.9 | 28.4 | 30.4 | 29.6 | 31.4 | 33.3 | 33.4 |
| SSIM↑ | .843 | .879 | .882 | .916 | .907 | .932 | .925 | .948 | .941 | .956 | .970 | .967 |
| LPIPS↓ | .249 | .161 | .174 | .097 | .129 | .073 | .098 | .051 | .072 | .039 | .031 | .032 |

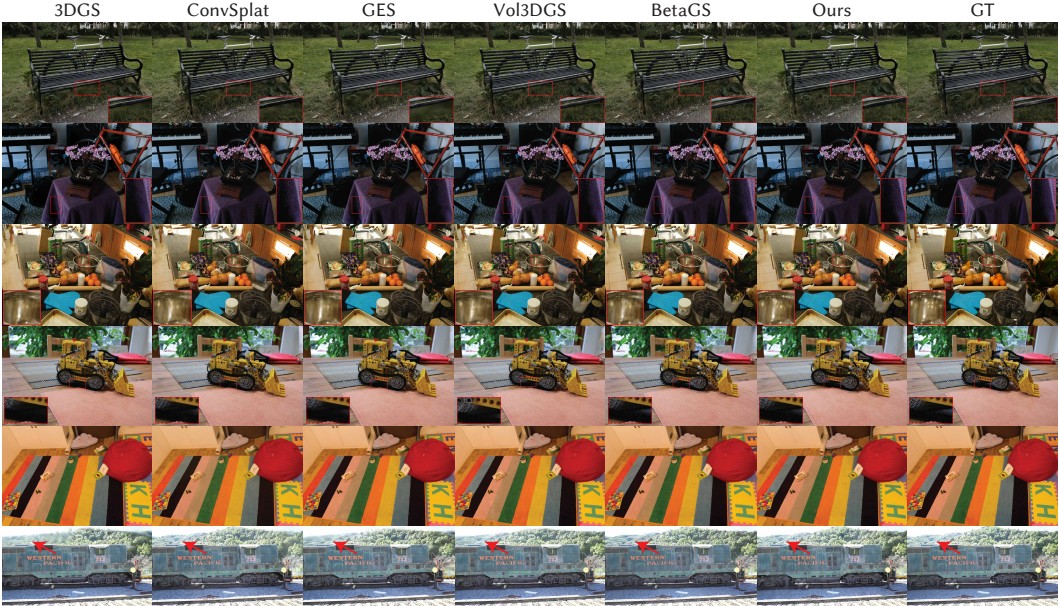

**Figure 6:** Visual comparison of our method against several primitive-based methods on the novel-view synthesis task for real scenes. We demonstrate that our neural primitives achieve high-fidelity results comparable to other approaches, requiring $10\times$ fewer primitives and $6\times$ fewer parameters.

### 4.1 PRIMITIVE EXPRESSIVITY

Leveraging a flexible neural density field and analytically exact integration, our method faithfully reproduces complex geometries with a small number of primitives. To demonstrate this, we optimize varying numbers of neural and Gaussian primitives to approximate the density fields of several 3D geometries from multiple views. We visualize both the renderings and the color-coded primitives in Fig. 4. We observe that a few neural primitives suffice to represent complex and diverse geometries, such as the teapot's curved handle, the smooth cut in the leaf, and the triangular leaf petiole. In contrast, Gaussian primitives are limited by their symmetric ellipsoidal shape and soft boundaries, making them unsuitable for accurately representing complex solid structures. For the toy examples of *teapot* and *leaf* in Fig. 4, neural primitives achieve superior performance while using $16\times$ fewer primitives and $4\times$ fewer parameters than 3DGS.

### 4.2 SYNTHETIC SCENES

**Protocol** We compare our method with 3DGS on the Synthetic NeRF dataset (Mildenhall et al., 2020) across varying memory budgets. Specifically, we resample the original meshes to target vertex counts (200, 500, 1k, 2k, 5k, 10k, 20k) and use them to initialize primitive positions for optimization, omitting primitive densification. We also include an "unlimited" setting, in which training follows the standard densification procedure with no primitive budget.

**Results** We report numerical results in Tab. 1. Our method outperforms 3DGS under limited memory budgets and achieves performance comparable to 3DGS when no memory constraints are imposed. Visual comparisons are shown in Fig. 5. For *Ficus*, a single primitive can already reconstruct

**Table 2:** Numerical comparisons on three real-scene datasets. For each method, we indicate whether it is splatting-based (Spl.) and/or neural (Neu.). We also report novel-view synthesis quality (PSNR↑, SSIM↑, LPIPS↓), rendering speed (in frames per second), and memory usage (in MB).

| | Spl. | Neu. | Mip-NeRF360 | | | | | Tanks & Temples | | | | | Deep Blending | | | | |
| | | | PSNR | SSIM | LPIPS | FPS | Mem | PSNR | SSIM | LPIPS | FPS | Mem | PSNR | SSIM | LPIPS | FPS | Mem |
|---|---|---|---|---|---|---|---|---|---|---|---|---|---|---|---|---|---|
| Plen | ✗ | ✗ | 23.08 | .626 | .463 | 7 | 2.1k | 21.08 | .719 | .379 | 13 | 2.3k | 23.06 | .795 | .510 | 11 | 2.7k |
| INGP | ✗ | ✓ | 25.59 | .699 | .331 | 9 | 48 | 21.92 | .745 | .305 | 14 | 48 | 24.96 | .817 | .390 | 3 | 48 |
| Mip360 | ✗ | ✓ | 27.69 | .792 | .237 | <1 | 9 | 22.22 | .759 | .257 | <1 | 9 | 29.40 | .901 | .245 | <1 | 9 |
| 3DGS | ✓ | ✗ | 27.21 | .815 | .214 | 152 | 734 | 23.14 | .841 | .183 | 188 | 411 | 29.41 | .903 | .243 | 154 | 676 |
| GES | ✓ | ✗ | 26.91 | .794 | .250 | 279 | 377 | 23.35 | .836 | .198 | 372 | 222 | 29.68 | .901 | .252 | 289 | 399 |
| BetaGS | ✓ | ✗ | 28.75 | .845 | .179 | 71 | 356 | 24.85 | .870 | .140 | 119 | 200 | 30.12 | .914 | .236 | 91 | 343 |
| ConvSplat | ✓ | ✗ | 26.66 | .769 | .266 | 103 | 77 | 23.71 | .842 | .170 | 83 | 83 | 29.61 | .901 | .245 | 66 | 110 |
| Vol3DGS | ✓ | ✗ | 27.30 | .813 | .209 | 124 | 703 | 23.74 | .854 | .167 | 168 | 255 | 29.72 | .908 | .247 | 156 | 844 |
| T-3DGS | ✓ | ✗ | 27.31 | .801 | .252 | 265 | 152 | 23.95 | .837 | .201 | 408 | 73 | 29.82 | .904 | .260 | 409 | 67 |
| **Ours** | ✓ | ✓ | 27.21 | .791 | .216 | 115 | 93 | 23.59 | .846 | .162 | 158 | 80 | 29.20 | .892 | .264 | 178 | 82 |

an entire leaf (highlighted by the white frame). Similarly, in *Lego*, neural primitives capture diverse geometries, such as the front shovel and the rear wheel. In contrast, Gaussian primitives perform poorly on these complex structures, particularly under tight budgets. We further analyze the impact of the primitive initialization strategy. Neural primitives with random initialization achieve an average PSNR of 33.36, which is comparable to 3DGS with random initialization (33.32) and to neural primitives with mesh-based initialization (33.40). This observation indicates that our representation behaves similarly to 3DGS across different initialization strategies.

## 4.3 REAL SCENES

**Protocol** For evaluation on real scenes, we follow established practice and use two scenes from Deep Blending (Hedman et al., 2018), two from Tanks & Temples (Knapitsch et al., 2017), and all scenes from the Mip-NeRF360 dataset (Barron et al., 2022). We compare against three method families: (i) splatting-based approaches with analytic primitives – 3DGS (Kerbl et al., 2023), GES (Hamdi et al., 2024), ConvSplat (Held et al., 2025), BetaGS (Liu et al., 2025), and Vol3DGS (Talegaonkar et al., 2025); (ii) T-3DGS (Mallick et al., 2024), which provides a more sophisticated mechanism for controlling the memory footprint of primitive-based representations; and (iii) monolithic representations – Plenoxels (Fridovich-Keil et al., 2022), INGP (Müller et al., 2022), and MipNeRF360 (Barron et al., 2022). All experiments use the official code released by the respective authors. Since our reproduced baseline results closely match those reported in the respective papers, we report the original numbers for consistency. For a fair comparison, all inference FPS values are measured on a single NVIDIA GeForce RTX 4090 GPU.

**Results** We summarize numerical results in Tab. 2. Our method achieves high-fidelity reconstructions with image quality and runtime comparable to state-of-the-art splatting-based approaches with analytic primitives, while generally requiring substantially less memory. Compared to monolithic neural representations, our neural splatting-based representation is more than an order of magnitude faster. While T-3DGS attains a similar trade-off, its control mechanisms are orthogonal to our contribution, which focuses on the representation itself; "taming" our neural primitives can be expected to yield significant gains as well. Fig. 6 confirms that our reconstructions are on par with the state of the art. In particular, our approach accurately captures fine-grained, structured geometry, such as the carpet region (highlighted in red) in the *Kitchen* and *Bonsai* scenes.

## 5 ABLATION STUDIES

Here, we first investigate an alternative neural integration strategy compatible with a neural representation (Sec. 5.1). We then analyze the impact of the key parameters in our model formulation (Sec. 5.2) as well as the effect of the geometry regularization term (Sec. 5.3).

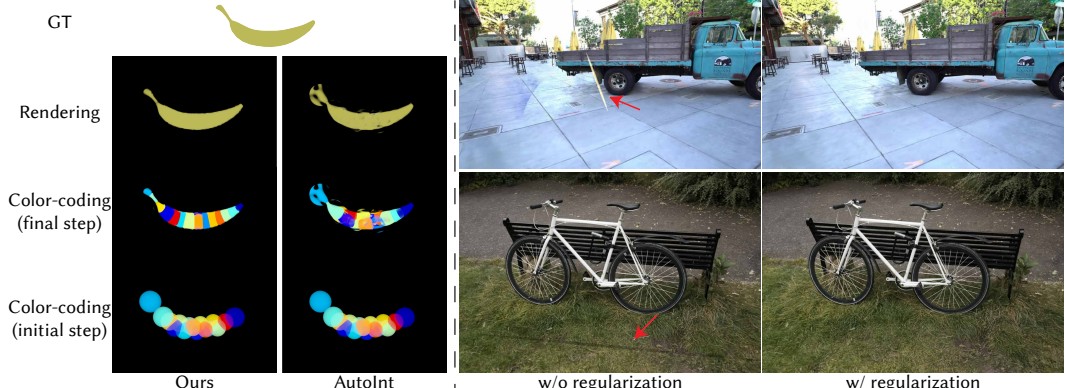

**Figure 7:** We analyze (left) the effect of an alternative neural integration strategy, AutoInt (Lindell et al., 2021), and (right) the effect of geometry regularization during training.

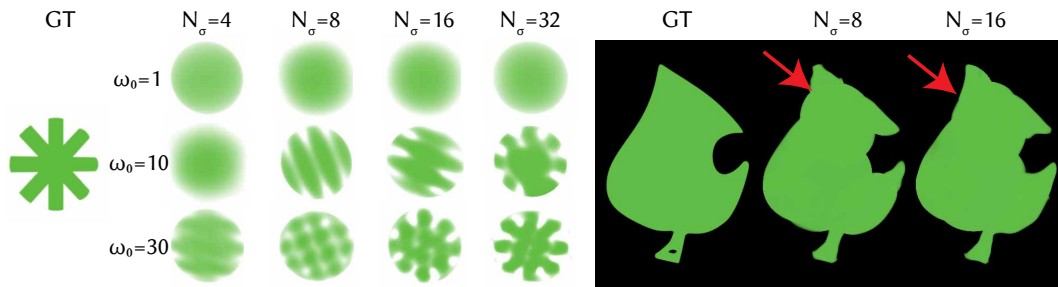

**Figure 8:** We visualize the effect of network configuration ($N_\sigma$ and $\omega_0$) on the expressivity of our neural density field.

## 5.1 NEURAL INTEGRATION

AutoInt (Lindell et al., 2021) is an alternative approach for computing line integrals in a neural field, which we compare in Fig. 7, left. AutoInt uses a ray-based parameterization and applies automatic differentiation with respect to ray depth during training to obtain an integral network. However, this induces view-dependent density, leading to inconsistencies across viewpoints. In contrast, our method models the density field with a shallow network that depends only on 3D position, ensuring multi-view consistency.

## 5.2 PRIMITIVE NETWORK CONFIGURATION

We analyze how the number of hidden neurons ($N_\sigma$) and the frequency multiplier ($\omega_0$) affect the expressivity of our neural representation. We vary $N_\sigma \in \{4, 8, 16, 32\}$ and $\omega_0 \in \{1, 10, 30\}$, and run experiments on *Snowflake* using a single primitive and *Leaf* using eight primitives. As shown in Fig. 8, larger $N_\sigma$ and higher $\omega_0$ better reproduce the *Snowflake* structure and the smooth contours of *Leaf*. Although $N_\sigma = 16$ offers greater expressivity than 8 in toy settings, this advantage diminishes on real scenes due to the difficulty of optimizing a highly under-constrained problem.

We further analyze the effect of the network configuration on all MipNeRF360 scenes. To facilitate a meaningful comparison, we resume training from a pretrained checkpoint and disable densification, thereby eliminating the influence of the number of primitives. We report PSNR values and memory footprint in Tab. 3. As $\omega_0$ increases from 0 to 50, the PSNR initially improves and reaches its peak for $\omega_0 = 30$, after which it slightly declines. At $\omega_0 = 30$, increasing $N_\sigma$ from 4 to 8 yields a larger quality improvement than increasing it from 8 to 16. However, the memory footprint from 8 to 16 is significantly higher (37 MB) than that from 4 to 8 (19 MB). Balancing memory footprint and expressivity, we set $N_\sigma = 8$ and $\omega_0 = 30$ as the default configuration for all experiments.

**Table 3:** Ablation study of network configuration ($N_\sigma$ and $\omega_0$) on all MipNeRF360 scenes. For each setup, we report the memory footprint (in MB) and image quality (PSNR).

| $N_\sigma$ | Mem | PSNR | | | |
|---|---|---|---|---|---|
| | | $\omega_0 = 1$ | $\omega_0 = 10$ | $\omega_0 = 30$ | $\omega_0 = 50$ |
| 4 | 73 | 26.35 | 27.29 | 27.33 | 27.12 |
| 8 | 92 | 26.32 | 27.42 | 27.54 | 27.41 |
| 16 | 129 | 26.19 | 27.46 | 27.62 | 27.55 |

## 5.3 GEOMETRY REGULARIZATION

Jointly optimizing millions of neural primitives in complex scenes is highly under-constrained and prone to local minima, often resulting in extreme geometries, as shown in Fig. 7, right. We find that geometric regularization stabilizes training by penalizing elongated primitives. While numerical results remain similar, the regularization yields clear qualitative improvements.

## 6 DISCUSSION AND CONCLUSION

Our method is a novel radiance field representation that reconciles the expressivity of neural representations with the efficiency of splatting-based rendering techniques. We identify accurate density field integration as a key factor for achieving high expressivity in novel-view synthesis. Inspired by neural radiance fields, we formulate each primitive as a shallow network, which enables exact integration by evaluating the analytical anti-derivative with only two queries, reducing the ray-marching burden. While such a network has comparably limited representational capacity, the primitive-based approach mitigates the limitation by enabling a collection of tiny primitives to jointly reconstruct fine-grained scene details. This design provides both computational accuracy and efficiency. Furthermore, we show that ellipsoid-bounded neural primitives can be integrated into a differentiable splatting-based renderer, achieving real-time rendering performance. Our experiments demonstrate that neural primitives produce high-fidelity results comparable to 3DGS while requiring 10× fewer primitives and 6× fewer parameters, and delivering 100× speedups over neural-based methods. We believe that *splattable neural representation* opens new possibilities of integrating neural-based representation with splatting-based rendering techniques.

Although neural primitives exhibit substantial expressivity with limited memory resources, the complexity of the optimization landscape for millions of networks occasionally introduces convergence difficulties, hindering the expressivity of neural representations and slowing down convergence. For instance, on the MipNeRF360 dataset, neural primitives require an average wall-clock training time of 2.5 hours, which is approximately 2.5× longer than 3DGS. A promising avenue for future research is to develop effective optimization or training strategies, such as the stochastic preconditioning technique (Ling et al., 2025), to fully unleash the expressivity of neural primitives and accelerate convergence. Moreover, our neural density field is independent of the improvement strategies for the color field and densification in 3DGS, making it theoretically compatible with such techniques. For example, the error-guided adaptive densification strategies (Rota Bulò et al., 2024; Mallick et al., 2024) could be readily adapted to our framework. Spatially-varying textured primitives (Chao et al., 2025) could also be integrated with the proposed neural density field. Exploring such integrations remains an interesting future research direction.

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

# A    INTEGRATION COST

**Table 4:** Comparison of GPU profiling metrics in Nsight Compute.

| Metric | 3DGS | Ours |
|---|---|---|
| Compute (SM) Throughput [%] | 84.52 | 80.02 |
| Memory Throughput [%] | 53.09 | 80.02 |
| Executed Instructions [inst] | 868,881,013 | 3,406,837,112 |

**Goal**   We conduct a comprehensive analysis of the computational overhead of neural and Gaussian primitives, from macro to micro-level evaluations on a single NVIDIA GeForce RTX 4090 GPU. We begin by evaluating the total rendering time under the same primitive budget, which provides the computational cost of the full rendering pipeline, including preprocessing, primitive sorting/tiling, integration, and alpha blending. We then analyze the computational cost in the render kernel function, which consists of the integration and blending processes. Finally, we focus on the integration step, decomposing it into the floating-point (FP) operation level.

**Protocol**   We perform computation analysis on a set of 100k primitives with random positions inside a volume and other random properties (MLP weights for neural primitives, opacities for gaussians) using our method and 3DGS, under the camera configurations of the synthetic NeRF dataset. We report FPS for total rendering time, similar to other 3DGS-based approaches. And we rely on Nsight Profiling to evaluate the render kernel function, providing compute and memory throughput, and executed instructions. We additionally manually decompose the integration process into floating-point (FP) operation level and report the FLOP number.

**Analysis – FPS**   Neural primitives achieve an FPS of 218.87, which is approximately $2.5\times$ slower than 3DGS (546.54). This gap is expected, as neural primitives require additional computation, including the MLP query, integration, and ray/ellipsoid intersection. In real scenes, our method needs $10\times$ fewer primitives, which balances computational complexity with the number of primitives, enabling real-time performance.

**Analysis – Nsight**   We report the Nsight profiling results by monitoring the render kernel function in Tab. 4, which consists of integration and alpha blending steps. Since our method and 3DGS employ the same alpha blending structure, the performance difference solely comes from the integration part. Both 3DGS and our method rely heavily on GPU compute ($84.52\%$ vs. $80.02\%$). Unlike 3DGS, which keeps its parameters in shared memory, our implementation must repeatedly access global memory for MLP weights, resulting in a higher memory throughput of $80.02\%$. The total number of executed instructions for neural primitives is $3.4 \times 10^9$, approximately four times higher than that of 3DGS ($8.7 \times 10^8$).

**Analysis – FLOP**   We further decompose the integration process and provide a FLOP-based analysis focused on the integration step. Using heuristic operation costs, where basic arithmetic, division, and exp/sin operations are assigned costs of 1, 4, and 30, respectively, 3DGS requires approximately 43 FLOPs per integration, whereas our method requires roughly 769 FLOPs.

**Summary**   Importantly, although neural primitives yield higher computational costs across FPS, Nsight, and FLOP-based analysis, our method requires roughly $10\times$ fewer primitives to represent the same scene as 3DGS. As a result, the overall rendering speed during inference remains over 100 FPS, which is comparable to 3DGS and ensures real-time performance.

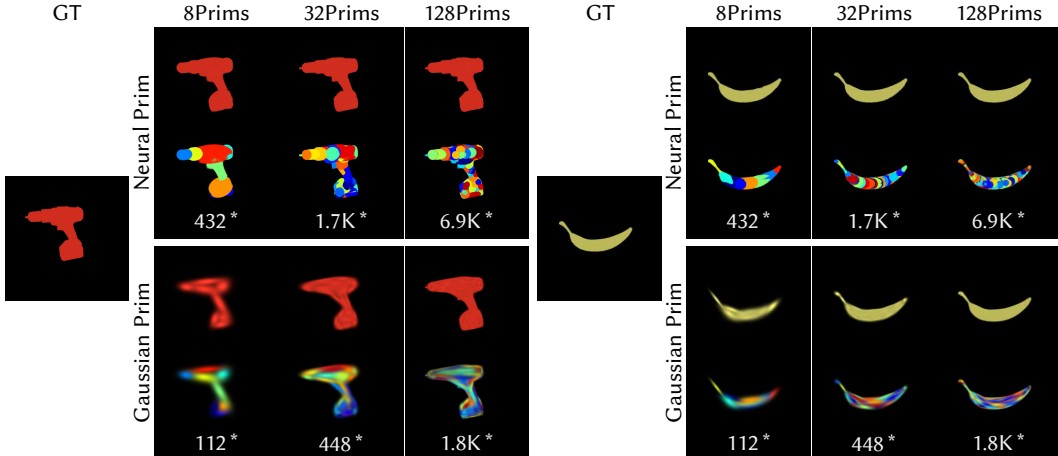

**Figure 9:** Demonstration of the expressivity of the proposed neural density field. We train both neural and Gaussian primitives on the *drill gun* and *banana* under different numbers of primitives. For each example, we visualize the reconstructed density field and color-coded primitives, illustrating how neural primitives are trained to represent complex structures. ∗ denotes the total number of parameters.

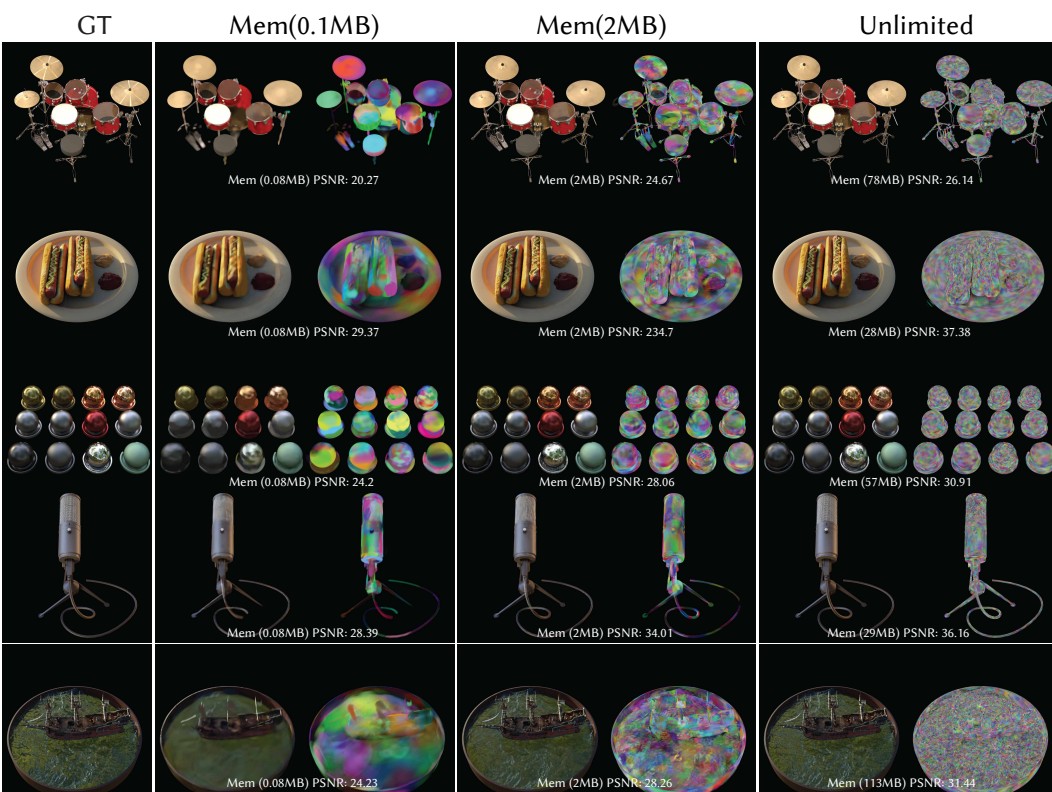

**Figure 10:** More visual results on the Synthetic NeRF dataset under limited and unlimited memory budgets.

## B  HYPERPARAMETERS

We train all scenes using the Adam optimizer (Kingma & Ba, 2015), with a learning rate of $10^{-3}$ for the MLP. For the datasets BlenderNeRF, Mip-NeRF360, Tanks & Temples, and Deep Blending, we adopt the following learning rate hyperparameters: primitive means ($1.6 \times 10^{-4}$), scales ($5 \times 10^{-3}$), quaternions ($10^{-3}$), and SH coefficients ($2.5 \times 10^{-3}$). Population control is governed by a growing scale threshold of $10^{-2}$ and a pruning scale threshold of $0.5$.

| 3DGS | ConvSplat | GES | Ours | GT |
|---|---|---|---|---|

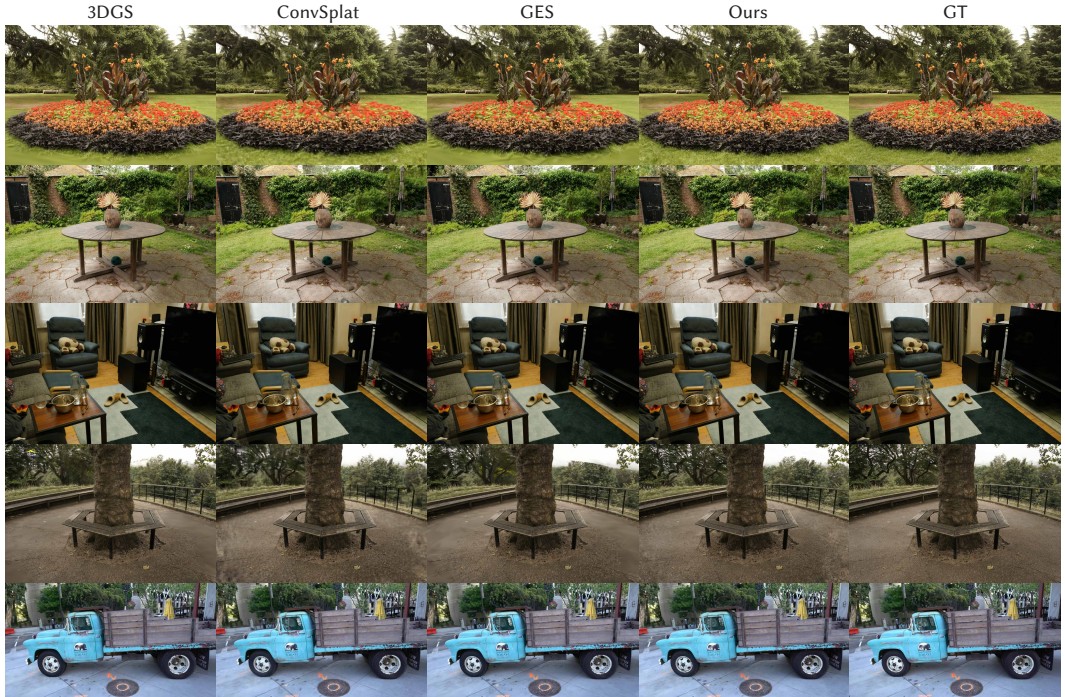

**Figure 11:** More visual comparison on real datasets.

**Table 5:** PSNR scores of each scene in the Blender Synthetic Dataset.

|  | Chair | Drums | Ficus | Hotdog | Lego | Materials | Mic | Ship |
|---|---|---|---|---|---|---|---|---|
| 3DGS 500 Prims | 22.91 | 19.09 | 22.39 | 25.55 | 20.99 | 22.99 | 27.59 | 23.32 |
| Ours 200 Prims | 24.60 | 20.27 | 23.87 | 29.37 | 22.56 | 24.20 | 28.39 | 24.23 |
| 3DGS $10k$ Prims | 29.19 | 23.56 | 29.68 | 32.31 | 26.72 | 27.28 | 31.19 | 27.03 |
| Ours $5k$ Prims | 31.24 | 24.67 | 32.86 | 34.71 | 29.33 | 28.06 | 34.01 | 28.26 |
| 3DGS nolimit | 35.83 | 26.15 | 34.87 | 37.72 | 35.78 | 30.00 | 35.36 | 30.80 |
| Ours nolimit | 34.58 | 26.13 | 34.81 | 37.38 | 35.41 | 30.91 | 36.16 | 31.44 |

For Mip-NeRF360, Tanks & Temples, and Deep Blending, MLP-gradient-based densification and pruning are performed every 500 iterations between 1k and 15k, using thresholds of $10^{-4}$ and $2 \times 10^{-6}$, respectively. In contrast, BlenderNeRF uses slightly lower thresholds ($10^{-5}$ and $10^{-6}$) and the densification routine is executed more frequently, every 200 iterations, starting at $1k$ and continuing until $20k$.

## C    MORE RESULTS

### C.1    NUMERICAL RESULTS

**BlenderNeRF Synthetic Dataset**    Our neural primitive requires 41 parameters from its 8-neuron MLP, 10 from geometry (3 for means, 3 scales, and 4 for quaternion), and 48 from SHs, in total 99 parameters, $1.68\times$ more than 3DGS parameters (59). We report per-scene image metrics (PSNR, LPIPS, and SSIM) in Tab. 5, Tab. 6, and Tab. 7, under different memory budgets. For a fair comparison, we constrain the number of primitives in our system to half that of 3DGS and report numerical results under this setting. As shown in these three tables, the first two double rows exhibit an apple-to-apple comparison between 3DGS and our method under the same memory budget. Neural primitives outperform Gaussian primitives consistently, highlighting the expressivity of our representation. The last double rows in the three tables evaluate the performance of the two representations under unlimited memory budgets.

**Table 6:** SSIM scores of each scene in the Blender Synthetic Dataset.

|  | Chair | Drums | Ficus | Hotdog | Lego | Materials | Mic | Ship |
|---|---|---|---|---|---|---|---|---|
| 3DGS 500 Prims | 0.8564 | 0.8065 | 0.8769 | 0.8873 | 0.7819 | 0.8489 | 0.9282 | 0.7638 |
| Ours 200 Prims | 0.8890 | 0.8608 | 0.8983 | 0.9391 | 0.8247 | 0.889 | 0.9439 | 0.7893 |
| 3DGS $10k$ | 0.9420 | 0.9124 | 0.9585 | 0.9578 | 0.8946 | 0.9242 | 0.9738 | 0.8393 |
| Ours $5k$ | 0.9635 | 0.9343 | 0.9777 | 0.9722 | 0.9373 | 0.0471 | 0.9866 | 0.8677 |
| 3DGS nolimit | 0.9878 | 0.9548 | 0.9870 | 0.9852 | 0.9820 | 0.9600 | 0.9927 | 0.9070 |
| Ours nolimit | 0.9856 | 0.9475 | 0.9841 | 0.9838 | 0.9808 | 0.9632 | 0.9910 | 0.8993 |

**Table 7:** LPIPS scores of each scene in the Blender Synthetic Dataset.

|  | Chair | Drums | Ficus | Hotdog | Lego | Materials | Mic | Ship |
|---|---|---|---|---|---|---|---|---|
| 3DGS 500 | 0.2247 | 0.310 | 0.150 | 0.219 | 0.309 | 0.256 | 0.147 | 0.383 |
| Ours 200 | 0.1359 | 0.220 | 0.089 | 0.093 | 0.200 | 0.1328 | 0.0968 | 0.3182 |
| 3DGS $10k$ | 0.0813 | 0.121 | 0.038 | 0.072 | 0.131 | 0.098 | 0.034 | 0.208 |
| Ours 500 | 0.0385 | 0.043 | 0.015 | 0.0293 | 0.0467 | 0.0472 | 0.0130 | 0.1453 |
| 3DGS nolimit | 0.010 | 0.037 | 0.011 | 0.020 | 0.017 | 0.038 | 0.006 | 0.109 |
| Ours nolimit | 0.014 | 0.047 | 0.016 | 0.020 | 0.017 | 0.032 | 0.008 | 0.100 |

**Real Datasets** Tables 8 and 9 present the per-scene performance metrics of our neural primitives, including PSNR, SSIM, LPIPS, the number of primitives, and the associated model memory footprint.

## C.2 VISUAL RESULTS

**Toy Examples** We provide additional visual comparisons of our method and 3DGS on two toy examples (*drill gun* and *banana*). As shown in Fig. 9, neural primitives exhibit significantly greater expressivity than 3DGS, achieving reconstructions with superior quality using fewer primitives and parameters. In contrast, 3DGS struggles to capture solid density fields, sharp edges, and smooth contours in both *drill gun* and *banana*.

**Synthetic and Real Results** In Fig. 10, we demonstrate additional visual results on the synthetic NeRF dataset, evaluated on the models optimized under varying memory budgets. Moreover, we provide comparison and visual results on additional real scenes in Fig. 11.

## D APPLICATIONS

In addition to the novel view synthesis task, we show that neural primitives can be readily adapted into other multimodal tasks, such as dynamic and relighting, by introducing an additional input channel to the density field or incorporating a neural color field.

### D.1 VOLUMETRIC DYNAMIC NOVEL VIEW SYNTHESIS

**Method** The zero-order Spherical Harmonics (SH) coefficient of each primitive is modeled as a function of time $\xi_t$, expressed as a summation of a polynomial function and a Fourier series, similar to Lin et al. (2024):

$$S(\xi_t) = S_0 + P_n(\xi_t) + F_l(\xi_t), \tag{10}$$

where $S_0$ denotes the zero-order SH coefficient. The polynomial function is defined as:

$$P_n(\xi_t) = \sum_i^n a_i \xi_t^i, \tag{11}$$

and the Fourier series component is given by:

$$F_l(\xi_t) = \sum_i^l \left( b_i \cos(i\xi_t) + c_i \sin(i\xi_t) \right), \tag{12}$$

**Table 8:** Novel view synthesis results in Mip-NeRF360 dataset

|  | Mip-NeRF360 | | | | | | | | | Avg |
|---|---|---|---|---|---|---|---|---|---|---|
|  | Bicycle | Bonsai | Counter | Flower | Garden | Kitchen | Room | Stump | Treehill |  |
| PSNR | 24.28 | 32.61 | 29.44 | 20.46 | 27.03 | 31.82 | 31.85 | 24.73 | 22.64 | 27.21 |
| SSIM | 0.7028 | 0.9467 | 0.9140 | 0.5497 | 0.8421 | 0.9291 | 0.9295 | 0.6862 | 0.6169 | 0.7907 |
| LPIPS | 0.2658 | 0.1540 | 0.1643 | 0.3390 | 0.1319 | 0.1112 | 0.1750 | 0.2780 | 0.3249 | 0.2160 |
| # Prims | $3 \times 10^5$ | $1.9 \times 10^5$ | $1.7 \times 10^5$ | $2.5 \times 10^5$ | $2.6 \times 10^5$ | $2.2 \times 10^5$ | $1.7 \times 10^5$ | $3.3 \times 10^5$ | $2.8 \times 10^5$ | $2.4 \times 10^5$ |
| Mem (MB) | 116.25 | 73.30 | 66.22 | 93.55 | 97.32 | 84.10 | 65.22 | 124.15 | 104.59 | 91.63 |

**Table 9:** Novel view synthesis results in Tank& Temple and Deep Blending datasets.

|  | Tank&Temple | | Avg | Deep Blending | | Avg |
|---|---|---|---|---|---|---|
|  | Train | Truck |  | Playroom | Drjohnson |  |
| PSNR | 21.98 | 25.19 | 23.58 | 29.62 | 28.78 | 29.29 |
| SSIM | 0.8175 | 0.8780 | 0.8478 | 0.8955 | 0.8886 | 0.8921 |
| LPIPS | 0.1864 | 0.1330 | 0.1597 | 0.2626 | 0.2661 | 0.2644 |
| # Prims | $2.2 \times 10^5$ | $2.0 \times 10^5$ | $2.1 \times 10^5$ | $1.6 \times 10^5$ | $2.7 \times 10^5$ | $2.2 \times 10^5$ |
| Mem (MB) | 84.66 | 74.43 | 79.55 | 60.90 | 102.74 | 81.82 |

where $a_i, b_i, c_i \in \mathbb{R}$. In our experiments, we set both $l$ and $n$ to 4.

We begin by uniformly sampling $100,000$ primitives within the volume's bounding box and train the system over $100,000$ iterations. The densification process starts at iteration $1,000$ and continues until iteration $30,000$, executed at intervals of 500 iterations. Similar to static scene configurations, all hyperparameters remain the same.

**Data setup** We evaluate our method in the dynamic volumetric novel view synthesis setting using a synthetic dataset, including four volumetric effects from JangaFX[1]. Each effect is recorded by 40 cameras on the upper hemisphere to capture temporal evolution, with 38 cameras for training and 2 for testing. The *Colorful Smoke* and *Ground Explosion* scenes contain 128 and 130 timesteps per camera, while *Dust Tornado* and *Smoke Fire* each have 100 timesteps per camera.

**Training** To reconstruct the temporal evolution of the volumetric effects, we adopt an Eulerian approach by incorporating an additional temporal variable $\xi_t \in [0, 1]$ for timestamp into our neural density field.

The temporally and spatially variant density field $\sigma(\mathbf{x}, \xi_t)$ now is:

$$f_\sigma(\mathbf{x}, \xi_t) = W_2 \cos(\omega_0 \cdot W_1(\mathbf{x}) + \xi_t \cdot W_t + \mathbf{b}_1) + \mathbf{b}_2 \tag{13}$$

where learnable temporal weight $W_t \in \mathbb{R}^{N_\sigma}$. Furthermore, the zero-order SH coefficients for each primitive are modeled as a function of time $\xi_t$ by expressing them as the sum of a polynomial function and a Fourier series.

**Results** Fig. 13 shows the visual results of our representation. By introducing an additional dimension, our system effectively captures the temporal evolution of the volumetric effects.

## D.2 RELIGHTING

**Method** Unlike novel view synthesis, where color is represented as SH coefficient, in relighting task, the primitive color per view ray is formulated as a combination of constant color $\mathbf{c}_{\text{dc}}$ and neural color function of 3D position $\mathbf{x}$, view direction $\mathbf{d}$ and light direction $\mathbf{d}_l$.

$$\mathbf{c} = \mathbf{c}_{\text{dc}} + \mathbf{c}(\mathbf{x}, \mathbf{d}, \mathbf{d}_l), \tag{14}$$

To smoothly adapt the relighting application to our neural representation, we incorporate a per-primitive color network field, where the light direction is computed relative to the center of each primitive.

---

[1]https://jangafx.com/software/embergen/download/free-vdb-animations

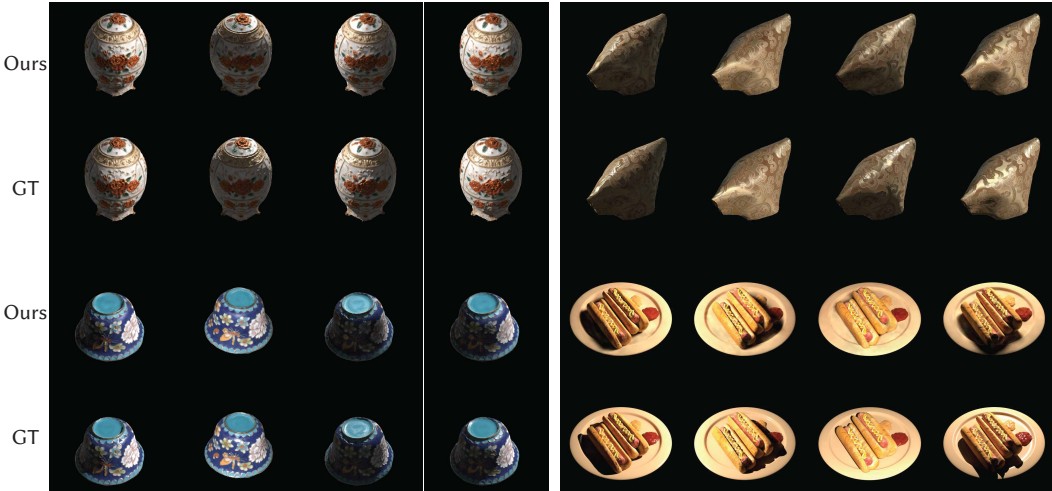

**Figure 12:** Relighting results under OLAT illumination using neural primitives.

**Dataset and Training Setup**    We conduct relighting application on One-Light-at-a-Time (OLAT) datasets provided by Bi et al. (2024); Kang et al. (2019), including: (1) rendered images of synthetic NeRF scenes, and (2) rendered images of real captures.

**Results**    We show relighting results in Fig. 12. Our relighting strategy can achieve decent image-based rendering without requiring intrinsic properties, capturing specular reflection (*fabrics* example provided in Fig. 12) and complex self-shadowing (refer to *hotdog* in Fig. 12 and *lego* scene in Fig. 1).

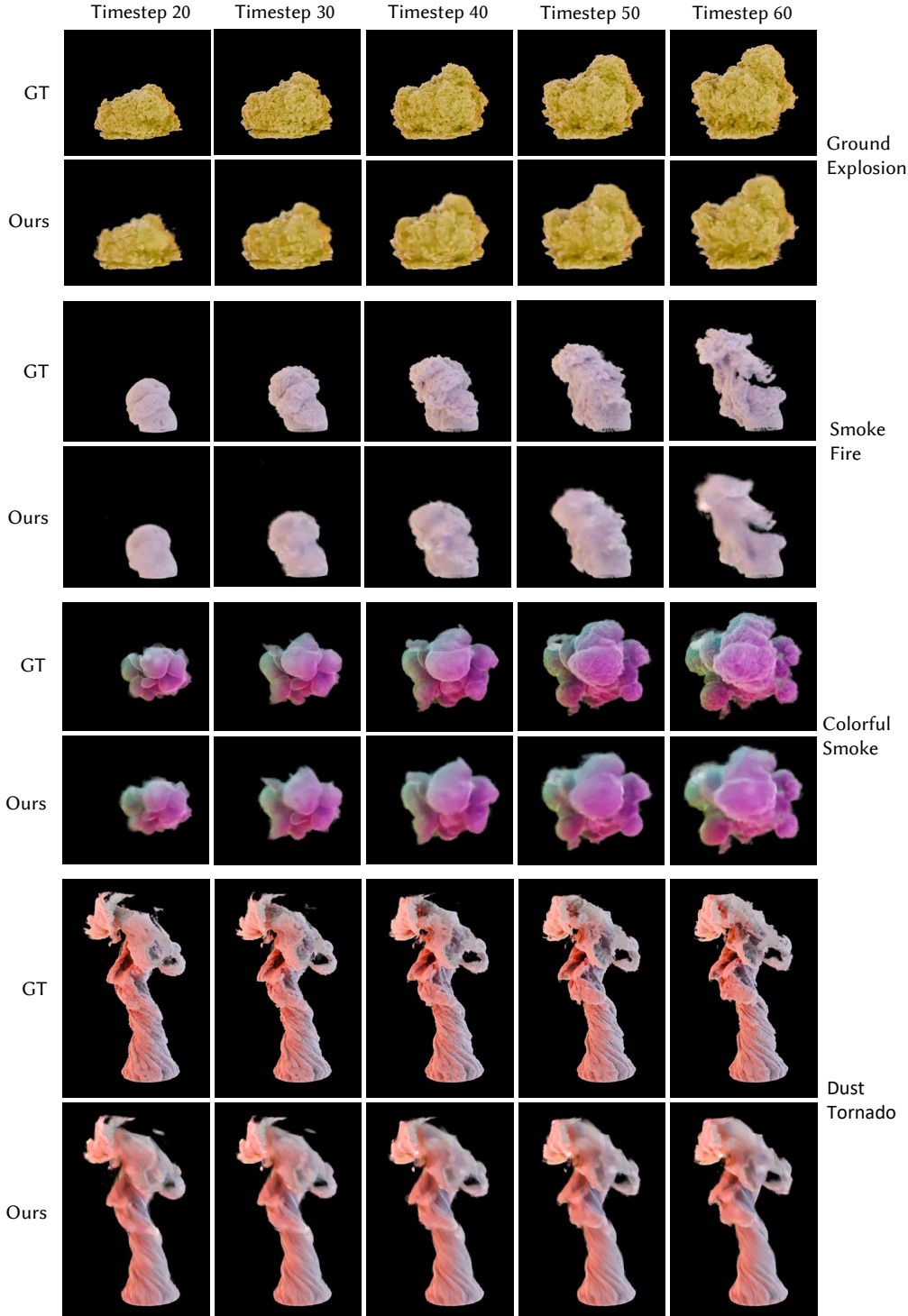

**Figure 13:** We demonstrate our results for volumetric dynamic view synthesis. By introducing an additional time dimension $\xi_t$, our neural primitives can reconstruct the scene's evolution and synthesize coherent results for volumetric dynamic scenes.

