# OpenReview forum: "Splat the Net: Radiance Fields with Splattable Neural Primitives"
_ICLR.cc/2026/Conference — ICLR 2026 Poster_

### Official Review · Reviewer_awhh · 2025-10-24

**Soundness:** 3
**Presentation:** 3
**Contribution:** 3
**Rating:** 8
**Confidence:** 4

**Summary:**

The paper proposes Splat the Net, a unified representation that integrates the expressivity of neural radiance fields with the real-time rendering efficiency of splatting-based approaches. Each primitive in the proposed framework defines a bounded neural density field represented by a shallow sinusoidal network, for which the authors derive a closed-form solution to line integrals along view rays.
This analytic construction practically allows accurate, differentiable rendering without ray marching. The approach achieves state-of-the-art visual quality on multiple benchmarks, such as NeRF Synthetic, Mip-NeRF360, and Tanks & Temples, while requiring an order of magnitude fewer primitives and parameters than 3D Gaussian Splatting.

**Strengths:**

1. The proposed framework is novel and well-explored. By introducing neural primitives that can be analytically integrated and splatted, the authors effectively bridge two traditionally separate families of radiance field models.
2. The resulting design reconciles neural expressivity with analytic efficiency, achieving a favorable trade-off between compactness, rendering quality, and runtime.
3. The paper provides extensive experiments across synthetic and real benchmarks, complemented by ablations and visual comparisons that clearly support the claims.

**Weaknesses:**

1. The reliance on per-primitive neural parameters may constrain the method’s scalability and practicality, particularly in memory-limited environments.
2. While the experimental results are comprehensive, the paper could be further strengthened by a brief discussion of scenarios where the proposed neural primitives may be less effective, which would help clarify the scope and robustness of the approach.

**Questions:**

Given that training is more computationally demanding than purely analytic splatting, are there potential strategies—such as improved initialization, parameter sharing, or adaptive optimization—that could help accelerate convergence?

---

> ### Author Response · Authors · 2025-11-21
>
> **Weakness 1**
>
> Please refer to the **Extension to Large-Scale Scenes** section in the common questions comment.
>
> **Weakness 2**
>
> Please refer to the **Failure Modes** section in the common questions comment.
>
> **Question**
>
> Although neural primitives demonstrate stronger expressivity than Gaussian primitives, we acknowledge that there remains room for improvement, particularly in accelerating convergence. Future research can investigate effective optimization strategies to smooth the loss landscape, such as the stochastic preconditioning technique or adaptive densification methods, to further reduce memory cost and unleash the expressivity of neural primitives.

---

> > ### Comment · Reviewer_awhh · 2025-11-21
> >
> > Thanks for the rebuttal. I appreciate the authors for the comments and I would maintain my positive rating.

---

### Official Review · Reviewer_ByAG · 2025-10-29

**Soundness:** 3
**Presentation:** 2
**Contribution:** 3
**Rating:** 6
**Confidence:** 4

**Summary:**

The paper proposes splattable neural primitives: volumetric, ellipsoid-bounded primitives whose density is represented by a shallow neural network with periodic activation. A key technical claim is a closed-form antiderivative for line integrals through the neural density, which yields a perspectively accurate splatting kernel and thus avoids ray marching while retaining neural expressivity. Empirically, on synthetic and real novel-view synthesis benchmarks, the method targets the quality/speed of 3D Gaussian Splatting (3DGS) while using ~10× fewer primitives and ~6× fewer parameters, attributing the gains to the representation itself rather than to heavy control frameworks.

**Strengths:**

This paper introduces a neural primitive whose volumetric density is learned via a one-hidden-layer MLP, yet remains analytically integrable along rays. This bridges the perceived “neural vs. primitive” dichotomy and is, to my knowledge, a first in making the primitive itself neural while still splatting.
The analytical formulation is clearly spelled out (ray–ellipsoid intersection, anti-derivative, front-to-back alpha compositing), and the implementation details include population control (split/duplicate/prune via weight-gradient magnitudes) and geometric regularization to avoid degenerate ellipsoids.
Ablations probe network width/frequency and the effect of regularization; comparisons to an alternative neural integration strategy (AutoInt) clarify multi-view consistency benefits.

**Weaknesses:**

1. While the representation is new, final image quality/speed sometimes appears comparable to strong modern 3DGS variants that incorporate compression/regularization/adaptive control (e.g., BetaGS, T-3DGS, structured/linear kernels), some of which achieve very low memory or high FPS. A more direct, apples-to-apples model-size / memory / bandwidth comparison to these compression-oriented pipelines would clarify the net practical advantage. (Table 2 partially covers this, but a focused compression study would help.)
2. Adding per-primitive neural components (albeit shallow) complicates implementation relative to purely analytic Gaussians, which map naturally to existing graphics pipelines and hardware rasterization paths. This may limit industrial adoption unless the benefits (fewer primitives, comparable speed) translate into easier deployment (e.g., on mobile/embedded) than a tuned Gaussian pipeline. A short discussion of engine integration, batching, and runtime kernels would strengthen practical significance. (Authors do note PyTorch/CUDA implementation.)
3. On the synthetic dataset, initializing primitive positions from resampled ground-truth meshes risks leaking geometry and may overstate robustness; a stronger setting would evaluate multiple non-oracle inits (sparse SfM points, random, noisy depth). The paper acknowledges slow convergence vs. Gaussians and extends training, which heightens the importance of init robustness.
4. It may be worth to consider augmenting each primitive with a small learnable feature vector as additional input to the MLP to further boost expressivity under fixed primitive counts; this could bridge to richer local modulation without exploding primitive numbers.

**Questions:**

How robust is training to different initializations (e.g., sparse SfM points, noisy/sparse seeds, or purely random placements/shapes)? Can you report quantitative results (PSNR/SSIM/LPIPS, convergence rate) and qualitative failure modes across several inits on both synthetic and real scenes? (This would mitigate concerns about mesh-based seeding.)
The authors extend training to 100k iterations due to slower convergence. Can you share wall-clock training time comparisons and memory bandwidth/throughput metrics vs. 3DGS?

---

> ### Author Response · Authors · 2025-11-21
>
> **Weakness 1 - Compression-based methods**
>
> We do not compare against these methods because our representation is orthogonal to compression strategies, meaning that such techniques could, in principle, be integrated with neural primitives. For example, the error-guided densification approach proposed in Taming-3DGS could be readily adapted to our framework. Exploring such integrations represents an interesting direction for future research. In this paper, we therefore focus on comparisons with different base representations, which we regard as the most appropriate apples-to-apples evaluation.
>
> **Weakness 2 - Fit to the graphics pipeline**
>
> We follow common practice in 3DGS and employ a dedicated **software** rasterization implementation based on CUDA. Owing to our main technical contribution, splatting neural primitives is as analytical as splatting Gaussians. While we demonstrate highly favorable trade-offs in reconstruction quality, rendering speed, and memory footprint, aspects related to industrial adoption, engine integration, and similar practical considerations are exciting but beyond the scope of this work, which focuses on introducing a fundamentally new scene representation.
>
> **Weakness 3** and **Question a & b**
>
> Please refer to the **Ablation Study** section in the common questions comment.
>
> **Question c - Wall-clock Discussion**
>
> We report the wall-clock time for MipNeRF360 scenes: 3DGS (1 hour) vs Ours (2.5 hours).
>
> **Weakness 4 - Per-primitive Feature**
>
> Since our framework already employs per-primitive neural networks, the additional per-primitive learned features could be absorbed into the network weights, and thus may not be necessary.
>
> To make it clear, let the learnable feature of each primitive be $\mathbf{f}$ with associated weight $\tilde{W}_1$. The first linear layer of the MLP for an input $\mathbf{x}$ then computes: $W_1 \mathbf{x} + \tilde{W}_1 \mathbf{f} + \mathbf{b}_1$.
> Here the constribution $\tilde{W}_1 \mathbf{f}$ can be absorbed into $\mathbf{b}_1$.

---

> > ### Comment · Reviewer_ByAG · 2025-11-26
> >
> > I appreciate the authors' rebuttal and will keep my rating.

---

### Official Review · Reviewer_FKH4 · 2025-10-29

**Soundness:** 3
**Presentation:** 3
**Contribution:** 3
**Rating:** 6
**Confidence:** 4

**Summary:**

This paper introduces splattable neural primitives, a hybrid radiance field representation designed to unify the expressivity of neural radiance fields with the efficiency of splatting-based rendering. Each primitive is represented as an ellipsoid-bounded neural density field, parameterized by a shallow sinusoidal network that admits a closed-form integral along view rays. This analytical formulation eliminates the need for expensive ray marching while retaining multi-view consistency. Experiments on both synthetic and real-world datasets demonstrate that the proposed approach achieves comparable or superior performance to 3D Gaussian Splatting (3DGS), requiring ten times fewer primitives and fewer parameters overall.

**Strengths:**

1. The proposed formulation is conceptually sound and well motivated. Representing volumetric primitives as shallow neural fields bounded by ellipsoids provides an elegant way to connect neural and analytic splatting methods under a single theoretical framework.

2. The analytical derivation of the antiderivative for the density field is mathematically consistent and efficiently implemented, offering a clear path to rendering without ray marching.

3. Experimental results indicate that the proposed neural primitives maintain strong image quality under strict memory constraints, remaining both compact and efficient.

**Weaknesses:**

1. The overall training process is more complex, requiring more iterations and careful convergence control due to the optimization of numerous small neural modules.

2. While the ablation studies illustrate the role of model parameters, a more detailed analysis of trade-offs between expressivity, convergence, and stability would have strengthened the paper’s argument.

3. The impact of network width or frequency choices on quality and efficiency remains underexplored in the ablation section.

**Questions:**

1. How does performance vary with different configurations of network width and frequency?

2. The scalability of the method for very large scenes is insufficiently discussed, how does runtime and memory usage scale with the number of primitives in such scenes?

---

> ### Author Response · Authors · 2025-11-21
>
> **Weakness 1**
>
> Please refer to the **Optimization Challenges & Complexity** section in the common questions comment.
>
> **Weakness 2 & 3** and **Question 1**
>
> Please refer to the **Ablation Study** section in the common questions comment.
>
> **Question 2**
>
> Please refer to the **Extension to Large-Scale Scenes** section in the common questions comment.

---

> ### Comment · Reviewer_FKH4 · 2025-11-26
>
> The rebuttal has addressed my concerns. I will keep my rating.

---

### Official Review · Reviewer_4maN · 2025-10-31

**Soundness:** 2
**Presentation:** 2
**Contribution:** 2
**Rating:** 4
**Confidence:** 2

**Summary:**

This paper introduces "splattable neural primitives," a novel radiance field representation that combines the expressivity of neural networks with the rendering efficiency of primitive-based splatting. Each primitive is an ellipsoid-bounded volume with a shallow neural network defining its density field, enabling exact analytical integration along view rays. The method achieves real-time rendering performance comparable to 3D Gaussian Splatting (3DGS) while using significantly fewer primitives (10×) and parameters (6×), demonstrating strong results on synthetic and real-world novel-view synthesis benchmarks.

**Strengths:**

1. Novel Representation: The proposal of fundamentally neural primitives with closed-form ray integration is a conceptually clean and innovative contribution. It successfully bridges the gap between expressive neural fields and efficient splatting-based rendering, a notable advance in the field.
2. Empirical Efficiency: The method demonstrates compelling practical benefits, matching 3DGS's quality and speed while drastically reducing primitive and parameter counts. This efficiency is directly attributed to the representation's inherent expressivity, not external control mechanisms.

**Weaknesses:**

1. Optimization Challenges: The paper acknowledges slower convergence and difficulties in optimizing millions of neural primitives due to a complex loss landscape. This suggests the method may be less robust or more sensitive to training configurations compared to established baselines like 3DGS.
2. Limited Ablation on Real Scenes: While toy examples (e.g., Snowflake, Leaf) effectively showcase expressivity, the ablation studies on network width and regularization lack depth for complex real-world scenes. The claimed expressivity advantage is not fully quantified or visually demonstrated on challenging benchmarks.

**Questions:**

1. Scalability & Robustness: Given the optimization difficulties mentioned, how does the method scale to extremely large, unbounded outdoor scenes? Are there specific types of scenes or geometries where the neural primitives consistently fail or underperform?
2. Integration Cost: The paper emphasizes "exact" and "efficient" integration. What is the precise computational overhead of evaluating the analytical anti-derivative compared to a single 3D Gaussian splatting kernel? A breakdown of rendering time (integration vs. blending) would clarify the practical trade-offs.

---

> ### Author Response · Authors · 2025-11-21
>
> **Weakness 1 - Optimization Challenges**
>
> Please refer to the **Optimization Challenges & Complexity** section in the common questions comment.
>
> **Weakness 2 - Ablation on real scenes**
>
> In Table 2, we show that neural primitives achieve comparable results to Gaussian primitives while using an order of magnitude fewer primitives and parameters. This provides strong evidence of the expressivity of neural primitives in challenging real-scene scenarios.
> For further ablation study in real scenes, please consult the **Ablation Study** section above, where we conduct experiments under different settings (base frequencies and different neurons) on the MipNeRF360 dataset.
> Regarding different benchmarks, we follow the community standard metrics for the novel view synthesis task that we tackle (PSNR, SSIM, LPIPS). These widely adopted metrics provide a fair comparison across the prior work.
>
> **Question 1 - Scalability & Robustness**
>
> Please refer to the **Extension to Large-Scale Scenes** and **Failure Modes** sections in the common questions comment.
>
>
> **Question 2 - Integration Cost**
>
> **Goal** We conduct a comprehensive analysis of the computational overhead of neural and Gaussian primitives, from macro to micro-level evaluations. We begin by evaluating the total rendering time under the same primitive budget, which provides the computational cost of the full rendering pipeline, including preprocessing, primitive sorting/tiling, integration, and alpha blending. We then analyze the computational cost in the render kernel function, which consists of the integration and blending processes. Finally, we focus on the integration step, decomposing it into the floating-point (FP) operation level.
>
> **Protocol** We perform computation analysis on a set of 100k primitives with random positions inside a volume and other random properties (MLP weights for neural primitives, opacities for gaussians) using our method and 3DGS, under the camera configurations of the synthetic NeRF dataset. We report FPS for total rendering time, similar to other 3DGS-based approaches. And we rely on Nsight Profiling to evaluate the render kernel function, providing compute and memory throughput, and executed instructions. We additionally manually decompose the integration process into floating-point (FP) operation level and report the FLOP number.
>
> **Analysis – FPS** Neural primitives achieve an FPS of 218.87, which is approximately $2.5×$ slower than 3DGS (546.54). This gap is expected, as neural primitives require additional computation, including the MLP query, integration, and ray/ellipsoid intersection. In real scenes, our method needs $10×$ fewer primitives, which balances computational complexity with the number of primitives, enabling real-time performance.
>
> **Analysis – Nsight** Then, we report the Nsight profiling results by monitoring the render kernel function, which consists of integration and alpha blending steps. Since our method and 3DGS employ the same alpha blending structure, the performance difference solely comes from the integration part. Both 3DGS and our method rely heavily on GPU compute (84.52% vs. 80.02%). Unlike 3DGS, which keeps its parameters in shared memory, our implementation must repeatedly access global memory for MLP weights, resulting in a higher memory throughput of 80.02%. The total number of executed instructions for neural primitives is $3.4×10^9$, approximately four times higher than the $8.7×10^8$ in 3DGS.
>
> | Metric | 3dgs | ours |
> |:------:|:----:|:----:|
> | Compute (SM) Throughput [%] | 84.52 | 80.02 |
> | Memory Throughput [%] | 53.09 | 80.02 |
> | Executed Instructions [inst] | 868,881,013 | 3,406,837,112 |
>
> **Analysis – FLOP** We further decompose the integration process and provide a FLOP-based analysis focused on the integration step. Using heuristic operation costs, where basic arithmetic, division, and exp/sin operations are assigned costs of 1, 4, and 30, respectively, 3DGS requires approximately 43 FLOPs per integration, whereas our method requires roughly 769 FLOPs.
>
> **Summary** Importantly, although neural primitives yield higher computational costs across FPS, Nsight, and FLOP-based analysis, our method requires roughly $10×$ fewer primitives to represent the same scene as 3DGS. As a result, the overall rendering speed during inference remains over 100 FPS, which is comparable to 3DGS and ensures real-time performance.

---

> > ### Comment · Reviewer_4maN · 2025-11-21
> >
> > Thanks for the rebuttal and reply. The authors have addressed my concerns, and I have upgraded the rating.

---

### Author Response · Authors · 2025-11-20
**Common Questions**

We thank all reviewers for their thoughtful comments and feedback. We will include all elaborations, analyses, and ablation studies discussed in this rebuttal in the final version of the paper. We first address the common questions raised by multiple reviewers, and then respond to the individual questions from each reviewer separately.
# **Extension to Large-Scale Scenes**
We agree that supporting very large-scale environments is an interesting aspect of scene representations. However, our focus in this work is on the core representational aspects and designing such a representation rather than on the engineering challenges associated with scaling to very large scenes, which we instead evaluate on standard datasets of the field. As the reviewers are aware, recent scene representations such as NeRF and 3DGS likewise addressed such settings only in later follow-up works [1,2], which have since developed into an active research direction of their own.

At the same time, we believe that our novel scene representation offers several characteristics that make it well-suited for scaling up. Requiring an order of magnitude fewer primitives and achieving a corresponding sixfold reduction in memory compared to 3DGS–while still maintaining over 100 FPS on challenging real scenes and without any dedicated control or compression mechanisms–makes splattable neural primitives a promising basis for future work in this direction. Since the number of primitives typically scales roughly linearly with scene size, starting from a representation that is already significantly more compact directly facilitates scaling to much larger environments. Moreover, our primitive-based representation can directly benefit from ongoing research on scaling up 3DGS, for example, through hierarchical structures.

[1]Grid-guided Neural Radiance Fields for Large Urban Scenes

[2]CityGaussian: Real-time High-quality Large-Scale Scene Rendering with Gaussians

# **Optimization Challenges & Complexity**
While the total number of parameters in our representation is significantly lower than in established 3DGS (Ours-92 MB vs 3DGS-734 MB), the optimization dynamics of our neural primitives differ and typically require more training iterations to converge. Nevertheless, we do not consider the optimization to be more complex than in 3DGS (based on real scene evaluation experiments). In fact, our approach relies on **fewer** stabilizing control mechanisms and requires tuning **fewer** hyperparameters–for instance, it does not rely on heuristic opacity resetting at regular intervals like 3DGS does. It remains an interesting and important future research direction to investigate methods for accelerating convergence and improving the optimization process of neural primitives.
# **Failure Modes**
While we occasionally observe minor difficulties of our representation in accurately fitting certain scene elements, we did not identify any discernible failure modes. These effects are not specific to particular shapes or scenes but tend to correlate with overall scene complexity. We will incorporate a more detailed discussion in a revised version of the paper to help readers better assess the challenging scenarios for our representation.
# **Ablation Study**
## *Network Width*
We analyze the effect of network width (number of neurons N and frequency w0) on all MipNeRF360 scenes. To facilitate a meaningful comparison, we resume training from a pretrained checkpoint and disable the densification process. This setup eliminates the influence of the number of primitives.
We report PSNR values and memory footprint in the following table:
|N|Memory(MB)|w0=1|w0=10|w0=30|w0=50|
|:-:|:-:|:-:|:-:|:-:|:-:|
|4|73|26.35|27.29|27.33|27.12|
|8|92|26.32|27.42|27.54|27.41|
|16|129|26.19|27.46|27.62|27.55|

We see that increasing the frequency​ improves PSNR up to w0=30. At w0=30, the PSNR gain from increasing N from 4 to 8 is larger than the gain from increasing N from 8 to 16. However, the memory cost from 8 to 16 is significantly higher (37MB) compared to the cost from 4 to 8 (19MB). Therefore, we set w0=30 and N=8 as our final configuration.
## *Initialization*
We appreciate the reviewer's suggestion for training robustness under different initialization strategies. However, we note that some of the suggested strategies (noisy/sparse seeds, or purely random placements/shapes) are not typically encountered in practice. For real scene evaluation, our initialization is the result of SfM (a point cloud), similar to other 3DGS-based methods. Regarding the NeRF Synthetic dataset, in the unlimited budget setting, we agree that mesh-based initialization, as employed in the limited budget setting for expressivity analysis, introduces ambiguities. We have now re-run the experiments with random initialization and observed that it performs the same (PSNR: ours 33.36 vs 3DGS 33.32). Hence, ours is not sensitive to initialization (ours achieves 33.40 PSNR with mesh-based initialization).

---

### Author Response · Authors · 2025-12-02

Dear Area Chairs,

We sincerely appreciate the thoughtful feedback and time of reviewers.

We have made every effort to address the questions and concerns reviewers raised. All reviewers have indicated that our responses have addressed their concerns. **Reviewer 4maN** has kindly increased their rating from 4 to 6, and the other three reviewers have maintained their original positive ratings.

We truly appreciate your time and consideration.

---

### Meta-Review · Area_Chair_6JMB · 2026-01-04

**Summary:**

The paper received mixed initial reviews with scores of 8, 6, 6, and 4. Reviewers broadly appreciated the proposed idea of splattable neural primitives, noting that embedding lightweight neural networks inside primitives offers a meaningful increase in expressivity over standard Gaussian splatting. The technical formulation was viewed as clear, and the empirical results on both synthetic and real-world benchmarks were considered promising. At the same time, some reviewers raised concerns about optimization stability, convergence speed, scalability to larger scenes, and the additional computational complexity introduced by neural primitives compared to classical 3DGS.

In the rebuttal, the authors provided detailed ablations and clarifications on network capacity, frequency choices, initialization robustness, and optimization behavior, along with thorough profiling of rendering and integration costs. These additions directly addressed the main technical concerns. One initially negative reviewer explicitly stated that their concerns were resolved and increased their score. The AC therefore anticipates a final score distribution of 8, 6, 6, 6 (see Reviewer Concerns and Reviewer Scores for details).

From the AC’s perspective, this work makes a solid and well-supported contribution by carefully exploring the trade-offs between expressivity and efficiency in radiance field representations. Beyond the well-discussed points regarding complexity, robustness, and scalability, the AC finds that the proposed model is particularly effective at modeling scenes with a reduced number of primitives, even though its peak reconstruction quality does not yet surpass that of recent 3DGS or other primitive-based variants. Overall, given the strong reviewer consensus, technical novelty, and contribution to the community, the AC recommends acceptance of this paper.

**Reviewer Concerns:**

### Reviewer 4maN (Score: 4)

- The reviewer initially raised concerns about optimization difficulty and convergence speed due to the use of neural networks inside each primitive, questioning training stability and scalability to larger scenes. They also felt that the expressive advantage of neural primitives was demonstrated primarily on toy examples, with insufficient evidence on complex real-world benchmarks, and requested clearer analysis of rendering and integration cost compared to Gaussian splatting.
- In the rebuttal, the authors provided extended ablations on network width and frequency on real benchmarks (e.g., Mip-NeRF360), detailed profiling of integration cost (including FPS, FLOPs, and Nsight analysis), and clarified scalability considerations. They also discussed optimization behavior and robustness in more detail in the common response.

---

### Reviewer FKH4 (Score: 6)

- The reviewer raised concerns about slower convergence and more complex training compared to Gaussian primitives, and asked for deeper discussion of the trade-offs between expressivity, optimization stability, and scalability. They also requested more analysis of network width and frequency choices.
- In response, the authors added ablations on network capacity and expanded discussion on optimization challenges and potential large-scale extensions.

---

### Reviewer ByAG (Score: 6)

- The reviewer questioned whether the reported gains were clearly superior to recent compressed or adaptive 3D Gaussian Splatting variants. Additional concerns included implementation complexity, robustness to different initialization strategies, lack of wall-clock training time comparisons, and a suggestion to include learnable feature vectors as additional network inputs.
- In the rebuttal, the authors clarified that compression methods are orthogonal and potentially complementary, discussed compatibility with graphics pipelines, and explained the relationship to additional learnable features. They also added experiments demonstrating robustness to different initialization strategies, clarified implementation aspects, and reported wall-clock training times for comparison with 3DGS.

---

### Reviewer awhh (Score: 8)

- The reviewer was largely positive but raised minor concerns about scalability and practical deployment, and requested discussion of failure cases or scenarios where the method may be less effective.
- In the rebuttal, the authors expanded the discussion of scalability limitations and acknowledged potential failure modes and directions for future improvement.

**Reviewer Scores:**

### Reviewer 4maN

- **Original score:** 4
- **Predicted final score:** 6
- **Rationale:** The reviewer explicitly stated that the rebuttal addressed their major concerns and indicated that they would like to upgrade their score accordingly.

---

### Reviewer FKH4

- **Original score:** 6
- **Predicted final score:** 6
- **Rationale:** The reviewer noted that their concerns were addressed in the rebuttal and indicated that they would maintain their positive score.

---

### Reviewer ByAG

- **Original score:** 6
- **Predicted final score:** 6
- **Rationale:** The reviewer appreciated the rebuttal and stated that they would keep their original positive score.

---

### Reviewer awhh

- **Original score:** 8
- **Predicted final score:** 8
- **Rationale:** The reviewer was already strongly positive, and the rebuttal addressed the minor points raised. They explicitly stated that they would keep their score.

---

### Decision · Program_Chairs · 2026-01-26

Accept (Poster)